# Optimization of the Enzymatic Protein Hydrolysis of By-Products from Seabream (*Sparus aurata*) and Seabass (*Dicentrarchus labrax*), Chemical and Functional Characterization

**DOI:** 10.3390/foods9101503

**Published:** 2020-10-20

**Authors:** Jesus Valcarcel, Noelia Sanz, José Antonio Vázquez

**Affiliations:** 1Marine Biotechnology and Bioprocesses Group, Marine Research Institute (IIM-CSIC), Eduardo Cabello 6, 36208 Vigo, Spain; nsanz@iim.csic.es (N.S.); jvazquez@iim.csic.es (J.A.V.); 2Recycling and Valorisation of Waste Materials Laboratory (REVAL), Marine Research Institute (IIM-CSIC), Eduardo Cabello 6, 36208 Vigo, Spain; 3Food Biochemistry Laboratory, Marine Research Institute (IIM-CSIC), Eduardo Cabello 6, 36208 Vigo, Spain

**Keywords:** seabream and seabass by-product valorization, *Sparus aurata*, *Dicentrarchus labrax*, fish protein hydrolysates, bioactive, mathematical optimization

## Abstract

Valorization of seabass and seabream by-products is becoming increasingly relevant, as marketing of these species moves from selling whole fish to filleting for convenience products. With this aim, we optimized for the first time the production of fish protein hydrolysates (FPH) by enzymatic hydrolysis from filleting by-products of these commercially relevant aquaculture species, isolating fish oil at the same time. On the whole, both fish yielded similar amounts of protein, but frames and trimmings (FT) were the best source, followed by heads and viscera. In vitro antioxidant and antihypertensive activities showed similar figures for both species, placing FPHs from FT as the most active. Molecular weights ranged from 1381 to 2023 Da, corresponding to the lowest values of FT, in line with the higher hydrolysis degrees observed. All FPHs reached high digestibility (>86%) and displayed an excellent amino acid profile in terms of essential amino acids and flavor, making them suitable as food additives and supplements.

## 1. Introduction

Global per capita fish consumption has more than doubled in the last fifty years, in line with increasing demand by a growing, wealthier, and more urbanized population. Up to the 1980s, demand was mainly met by increased wild capture, but aquaculture has rapidly closed the gap, surpassing consumption from fisheries in 2015 [1].

In the aquaculture landscape, species of seabass and seabream rank amongst the widest cultured species, reaching a global production in 2017 of more than 335,000 and 312,000 tons respectively. These figures represent around 15% (bass) and 14% (seabream) of the total marine fish cultured worldwide [2]. Historically, seabass and seabream have been marketed in Europe as whole fish. However, growing interest by consumers in convenience products has produced an increase in fillet production [3]. By-products of seabream and seabass account for around 60% and 50% respectively of total weight [4,5,6], therefore a significant amount of processing waste can be expected in the near future.

This biomass needs to be handled to avoid environmental problems, increasing at the same time resource efficiency to contribute to a sustainable fish supply. Seabass and seabream by-products contain significant amounts of high-quality protein rich in essential amino acids, and fat high in polyunsaturated fatty acids [6,7], making them suitable substrates for valorization. In particular, the application of proteases allows one to disgregate the material into a solid phase rich in mineral, an organic phase rich in fat and an aqueous phase with a high content of soluble protein, peptides and free amino acids [8,9]. In other species, these fish protein hydrolysates (FPHs) have demonstrated antioxidant, antihypertensive, anti-inflammatory, anti-microbial and anti-diabetic activities [10,11,12], as well as physicochemical properties suitable for food formulation [13].

Surprisingly, very few works have studied FPHs from seabream and seabass, possibly because of the limited availability of by-products so far. In seabream, the hydrolysis of gelatin obtained from fish scales released peptides with inhibitory activity against angiotensin I converting enzyme (ACE) [14], capable of reducing blood pressure in in vivo tests with rats [15]. FPHs from heads of seabream and seabass showed antioxidant activity, along with emulsifying and foaming properties [16,17], useful in improving the texture of fish mince [16].

The present study aims at determining the feasibility of FPH production from seabass and seabream by-products for food applications as a valorization strategy in aquaculture. We will first establish the optimal proteolysis conditions in fish heads, then upscale hydrolysis to viscera, trimmings and frames, while evaluating reaction kinetics. Quality and characteristics of the resulting products include amino acids analysis, molecular weight and peptide size distribution of FPHs, as well as antioxidant and anti-hypertensive activities.

## 2. Materials and Methods

### 2.1. Seabream and Seabass By-Products

Gilt-head seabream (*Sparus aurata*, Sb) and European seabass (*Dicentrarchus labrax*, Sbass) from aquaculture were acquired from food markets in Vigo (Spain). After filleting, heads (He), viscera (Vis), and frames along with trimmings (FT) were ground in a meat mincer and stored at −18 °C until use.

### 2.2. Optimization of Enzyme Hydrolysis

As a starting point, the combined influence of *pH* and temperature (*T*) on the hydrolysis of Sb and Sbass heads with Alcalase 2.4 L (2.4 AnsonUnit/g, AU/g enzyme, Nordisk, Bagsvaerd, Denmark) was evaluated by means of rotatable second order designs using five replicates at the center of the experimental domain [18]. Alcalase concentration, solid to liquid (S:L) ratio, and agitation speed were left invariable in these experiments (Appendix A). The responses (*Y*) evaluated were maximum hydrolysis (*H_m_*), concentration of soluble protein (Prs), and yield of digestion (*V_dig_*). After application of an orthogonal least-squares method the following polynomial equations were obtained, relating the effect of the independent variables on the responses:(1)$$Y=b_{0}+\sum_{i=1}^{n}b_{i}X_{i}+\sum_{\underset{j>i}{i=1}}^{n-1}\sum_{j=2}^{n}b_{ij}X_{i}X_{j}+\sum_{i=1}^{n}b_{ii}X_{i}^{2}$$
where: *Y* is the response tested, *b_0_* is the constant coefficient, *b_i_* is the coefficient of linear effect, *b_ij_* is the coefficient of combined effect, *b_ii_* is the coefficient of quadratic effect, *n* is the number of variables and *X_i_* and *X_j_* are the independent variables considered in each case. Statistical significance of coefficients was ascertained using Student’s t-test (α = 0.05) and goodness-of-fit using the coefficients of determination ($R_{2}$) and adjusted coefficients of determination ($R_{adj}^{2}$). Model consistency was ensured calculating mean square ratios from the Fisher F test (α = 0.05): *F1* = Model/Total error. The model is acceptable when: (i) $1\geq F_{den}^{num}$; and *F2* = (Model + Lack of fitting)/Model, (ii)$\;F2\leq F_{den}^{num}$. $F_{den}^{num}$ are the theoretical values (α = 0.05) with the corresponding degrees of freedom for numerator (num) and denominator (den). The experiments were carried out in a 100 mL glass reactor working as a pH-Stat system, with control of temperature, agitation and addition of reagents. The reaction was stopped by heating at 90 °C for 15 min.

Then, the effect of enzyme concentration on proteolysis of Sb_He and Sbass_He was assessed, maintaining the other experimental conditions constant (pH and T set to the values previously optimized). After 3 h of reaction, the content of the reactors was centrifuged (20 min, 15,000× *g*) and supernatants and precipitates were quantified.

### 2.3. Production of Enzymatic Hydrolysates

Production of FPHs was upscaled to 5 L glass reactors (pH-Stat system with control of temperature, agitation and addition of reagents), mixing 1 kg of minced by-products with 1 L of distilled water (solid:liquid ratio of (1:1)), using 5 M NaOH for pH-control. The reaction was performed at the optimal values estimated in the previous section. After 3 h of digestion the contents of the reactor were filtered (100 μm) to separate bones, followed by centrifugation (15,000× *g* for 20 min) and decantation for 15 min of the liquid phase to separate fish oil from FPHs. Finally, the enzyme was inactivated at 90 °C for 15 min.

Hydrolysis degree (*H*) was calculated as % according to the pH-Stat method [19] with mathematical modifications [20]. The time evolution of *H* was fitted to the Weibull equation [21]:(2)$$H=H_{m}\left\{1-\exp\left[-\ln2{\left(\frac{t}{\tau }\right)}^{\beta }\right]\right\}\;\mathrm{with}\;v_{m}=\frac{\beta H_{m}\ln2}{2\tau }\mathrm{with}v_{m}=\frac{\beta H_{m}\ln2}{2\tau }$$
where, *H* is the hydrolysis degree (%); *t* is the hydrolysis time (min); *H_m_* is the maximum hydrolysis degree (%); *β* is a dimensionless parameter related to the slope of the hydrolysis reaction; *v_m_* is the maximum hydrolysis rate (% min^−1^) and *τ* is the time required to reach the semi-maximum hydrolysis degree (min). The yield of digestion/liquefaction (V_dig_) of the raw by-products was also calculated (in %) [21].

### 2.4. Chemical and Biological Analyses of Substrates and Bioproducts Obtained

An initial assessment of by-product composition was carried out by determination of moisture, organic matter and ash percentage [22], total protein as total nitrogen × 6.25 [22], and total lipids [23]. The fatty acid profile in fish oil was quantified by gas chromatography after chemical methylation [24]. The hydrolysates were subjected to the following analyses: Total sugars [25]; total soluble protein [26]; total protein (total nitrogen × 6.25) [22]; amino acids by reaction with ninhydrin [27] using an amino acid analyser (Biochrom 30 series, Biochrom Ltd., Cambridge, UK); and in vitro digestibility (pepsin method: AOAC Official Method 971.09) according to the modifications suggested by Miller et al. [28].

The molecular weight distributions of FPHs were obtained by Gel Permeation Chromatography (GPC). The system (Agilent 1260 HPLC) consisted of a quaternary pump, injector, column oven, and refractive index, diode array and dual-angle light scattering detectors. The samples were eluted with 0.15 M ammonium acetate/0.2 M acetic acid (pH 4.5) at 1 mL/min after a 100 µL injection. Separation was achieved with a set of four Proteema columns (PSS, Germany): precolumn (5 µm, 8 × 50 mm), 30 Å (5 µm, 8 × 300 mm), 100 Å (5 µm, 8 × 300 mm), and 1000 Å (5 µm, 8 × 300 mm) kept at 30 °C. Detectors were calibrated with a polyethylene oxide standard of average molecular weight of 106 kDa (polydispersity index 1.05) from PSS (Mainz, Germany). Absolute molecular weight estimation was made with refractive index increments (dn/dc) of 0.185. In the case of peptides with molecular weight below 10 kDa, the samples of FPHs, after centrifugation on an Amicon-10 kDa centrifugal filter (MerckMillipore, Darmstadt, Germany), were quantified by HPLC (UV-detection at 220 nm) using a Superdex peptide 10/300 GL column (GE Healthcare Life Sciences, Little Chalfont, UK) with 0.1% trifluoroacetic acid in 30% acetonitrile as mobile phase. The flow rate was set to 0.4 mL/min and the column oven kept at 25 °C. The standards were Blue Dextran (2 MDa), Cytochrome c (12.4 kDa), Aprotinin (6.5 kDa), Angiotensin II (1046 Da), Leucine encephalin (555 Da), Val-Tyr-Val (379 Da) and Gly-Gln (221 Da).

Antioxidant (AO) activities in the hydrolysates were determined at the microplate scale: (i) 1,1-Diphenyl-2-picryhydrazyl (DPPH) radical-scavenging ability [29]; (ii) Crocin bleaching assay [30]; (iii) ABTS (2,2′-azinobis-(3-ethyl-benzothiazoline-6-sulphonic acid) bleaching method [29]. Antihypertensive (AH) activity was assessed by the in vitro angiotensin I converting enzyme (ACE) inhibitory activity (*I_ACE_*) assay [31], and the *IC*_50_ values (hydrolysate concentration that produces 50% of maximum *I_ACE_*) calculated by dose-response modelling [32]. AH and AO determinations were made in triplicate at 1 g/L of soluble protein.

### 2.5. Numerical and Statistical Analyses

Data fitting to mathematical equations and estimation of parameters was carried out with the non-linear least-squares (quasi-Newton) method, part of the macro ‘Solver’ (Microsoft Excel spreadsheet). Confidence intervals of parameter estimations (Student’s *t* test) and robustness of equations (Fisher’s F test) were determined with the “SolverAid” macro.

## 3. Results and Discussion

Filleting of seabass and seabream produced a significant quantity of by-products, as recovery of flesh only amounted to around half of the total fresh fish weight (52.9% for seabream and 47.4% for seabass). The discarded biomass consisted of heads (He), frames and trimmings (FT), and viscera (Vis), representing respectively 18.7%, 20.4%, and 8% of total fish weight for seabream, and 16.7%, 26.9%, and 9.1% for seabass. Each of these materials contained around 30% of organic matter, mainly composed of lipids and protein (Table 1). As expected for fatty fish, both species showed high lipid content, reaching 54.2% of dried weight in seabass viscera, but also contained a remarkable amount of protein (up to 52.3% in seabream heads). Both lipids and protein represent valuable compounds suitable for recovery through valorization processes.

Bearing this in mind, a process to solubilize protein by enzymatic hydrolysis was devised, establishing the optimal hydrolysis conditions in seabass and seabream heads as a first step. Then, these conditions were applied to trimmings and frames and viscera, releasing and separating at the same time the lipid fraction. Finally, the merits of this approach are discussed based on the composition of both fractions, and antioxidant and antihypertensive activities of the protein fraction.

### 3.1. Optimization of Proteolysis and FPH Production

As a starting point, the effect of pH and temperature on the proteolysis of seabass and seabream heads was studied using Alcalase as the proteolytic enzyme. This enzyme has proven versatile and efficient, capable of breaking down protein in a range of marine substrates such as shark cartilage [33], squid pens [34], salmon and tuna heads [35,36], or cod viscera [37].

The response surface methodology was applied to a two-variable factorial design (pH and temperature), maintaining constant agitation, solid to liquid ratio, enzyme concentration, and reaction time (Appendix A). Temperature and pH values that maximize hydrolysis responses were quite similar for both species, ranging from 56.4 to 60.7 °C and 8.0 to 8.7 (Table 2, response surfaces in Figure 1). The responses themselves (*Y_max_*), i.e., maximum hydrolysis, yield of digestion, and concentration of soluble protein, also showed close figures. The optima values were numerically derived from the polynomial equations obtained. Such equations correlated well with the experimental data as indicated by the adjusted correlation coefficients, ranging from 0.781 to 0.885, and were statistically robust, satisfying F-Fisher tests (data not shown). 

Then, the effect of Alcalase concentration on hydrolysis was assessed using the mean temperature and pH calculated from optima values of the three hydrolysis responses (Table 2). These were respectively set to 57.13 °C and 8.17 for seabream and 58.43 °C and 8.46 for seabass, keeping the rest of the conditions constant as in the first optimization experiment. The behaviour of maximum hydrolysis, yield of digestion and soluble protein were again similar for both seabass and seabream (histograms in Figure 1). Alcalase concentration produced no statistically significant effects (*p* > 0.05) on yield of digestion. The other two responses increased as enzyme concentration doubled from 0.1 to 0.2%, but rising concentration to 0.5% resulted in no significant differences (*p* > 0.05) with 0.2% protease. Consequently, a concentration of 0.2% Alcalase was chosen for the ensuing experiments.

Once the optimal proteolysis conditions were established for seabream and seabass heads, the process was upscaled to 5 L reactors and applied to heads, trimmings and viscera of both species. At the end of the enzymatic reaction, a solid fraction (mainly bones) was separated from the raw FPH (Figure 2). Subsequent centrifugation of this liquid phase allowed the recovery of fish oil after separation of the aqueous phase by decanting, which underwent thermal inactivation and drying to result in the final FPH product.

Quantification of each resulting fraction allowed us to identify differences between both species and the best sources of protein and lipids (Table 3). The total amount of protein recovered as a percentage of fresh fish weight was similar in both species, considering the sum of all three tissues. FPHs from the same tissue also showed little difference, with highest values found in frames and trimmings, followed by heads (around 50% lower) and viscera accounting for only 15% of the maximum. Considering previous experiments with the same solid to liquid ratio, soluble protein of frames and trimmings of seabass (73.0 g/L) and seabream (81.2 g/L) lies at the high end, compared with reported concentrations of 53.6 g/L in trout, 69.7 g/L in salmon [38] and 73.9 g/L in turbot [12].

The higher fat content in seabass determined in the raw by-products (Table 1) concurs with the amount of oil recovered (Table 3), more than doubling oil mass from seabream (considering the sum of all three tissues and their proportions based on fresh fish), accompanied by a higher amount of bones in seabream. Surprisingly, viscera of the latter contained more than 7% of bones, possibly remains of undigested feed. Despite the remarkable oil percentage of seabass viscera (27.5%), frames and trimmings rose as the richest source, as viscera only accounted for 9.1% of the whole seabass weight, while frames and trimmings reached 26.9%. Yield of digestion and digestibility were also similar between both species, with highest yield in viscera and digestibility greater than 86.3% in all samples. Sugar content was consistently low across tissues.

As a final step, hydrolysis kinetics for each material was modeled (Figure 3). In all cases the model fitted well the experimental data, as shown by correlation coefficients, from 0.928 to 0.997, and *p*-values lower than 0.0005 (data not shown). The model estimated the highest maximum hydrolysis degrees for frames and trimmings (21.5% in seabream and 21.7% in seabass), slightly lower values for heads (18.3% and 17.4% respectively), and remarkably different figures for viscera (19.27% in seabream vs. 12.96% in seabass). Previous works have reported higher hydrolysis degrees, reaching 36.3% for seabream heads and 38.5% and 40.5% in seabass heads [16,17]. This discrepancy may be partially explained by differences in methodology, as in the latter studies heads were boiled prior to FPH production and hydrolysis degree was expressed in terms of L-leucine, which leads to overestimation. Furthermore, maximum reaction rates in viscera almost doubled in seabass and were four times higher in seabream than in other tissues. As a result of these differences in *v_m_*, the reaction in viscera reached maximum hydrolysis long before the 3 h of the experiments, allowing for shorter reaction times (Figure 3).

### 3.2. Chemical Characterization

Beyond protein yield, the chemical characteristics of FPHs greatly influence their suitability as food and feed ingredients and hence their value, where the amino acids profile and molecular weight distributions are especially relevant.

Results of amino acid analysis ranked glutamic acid as the prevalent compound in all samples, followed by aspartic acid and glycine, and alanine, leucine and lysine, with small variations among tissues (Table 4). Previous works have also identified glutamic acid, aspartic acid and glycine as the major amino acids in FPHs from seabass and seabream heads [17], as well as in other fish [39]. Essential amino acids for humans comprise between 40 and 45% of the total amino acid content, higher than FPHs from other species such as trout (37%), salmon (33–36%), or turbot (28–31%), and similar to monkfish (40–42%) [12,38,40]. 

Characteristic fish flavor in marine species has been associated with Glu, Gly and Ala content, with Glu and Asp in particular contributing to the desirable umami taste [41]. In the FPHs analyzed, the sum of Glu, Gly and Ala varies from 28.3 to 32.1%, and Glu + Asp content from 22.4 to 25.9% (Table 4). Furthermore, Phe content, of bitter flavor, only reached 3.98 to 4.96%. While sensory analysis would be required, it can be anticipated that seabass and seabream FPHs could be potential ingredients in foods.

The molecular weight distributions evaluated by GPC appeared at long retention times, indicating considerable proteolysis. These were characterized by several overlapping peaks in the refractive index and UV detectors and a broad signal in light scattering. The average molecular weight (Mw) of this mixture of peptides ranged from 1381 to 2023 Da (Figure 4), where frames and trimmings were the lowest values. This was consistent with the higher hydrolysis degrees observed in this material (Figure 3).

The information provided by GPC with light scattering detection was complemented with HPLC analysis capable of separating the lowest end of the molecular weight distribution (Figure 5). The combination of both techniques allowed us to segregate the peptide distribution into molecular weight ranges (Table 5). In all species, the majority of peptides fell below 3 kDa (87.5 to 95.4%), and within this group a considerable fraction of the FPH samples (17.6 to 28.5%) displayed a molecular weight below 200 Da, indicating protein cleavage to individual amino acids or very short chains (2–3 units). For most samples, the highest proportion of peptides ranged from 1–3 kDa, except in seabream viscera and seabass frames and trimmings with slightly higher percentages from 0.2 to 1 kDa.

The lipidic fraction was also profiled to assess the quality of the fish oil recovered. Oleic, linoleic and palmitic acids dominated fatty acid composition in all cases, accounting from 65.2 to 69.9% of the total (Appendix A). EPA and DHA content was rather modest, ranging from 2.76 to 4.8% and from 4.82 to 10.72% respectively (7.57 to 15.51% combined), when compared to wild-capture whole fish (11.3–27.6% EPA + DHA) [39]. However, the values obtained here fare well in comparison to other aquaculture reared species such as trout (2.8% EPA + DHA) [38], salmon (3.2–3.7% EPA + DHA) [38] or turbot (4.3–5.7% EPA + DHA) [12]. The highest EPA + DHA proportion occurred in seabream heads (15.5%) and heads and trimmings (14.0%). Omega 6 predominated over omega 3 fatty acids, with higher ω-3/ω-6 ratios in seabream (from 0.76 to 0.83) than in seabass (0.51 to 0.89). In other aquaculture species, ratios below 1 are common [12,38], again in opposition to wild fish with ratios between 2.9 and 9.4 [39].

### 3.3. Antioxidant and Antihypertensive Activities of FPHs

All three antioxidant assays showed similar patterns, with comparable results for the same tissue in each species. A previous study found differences in antioxidant capacity between FPHs from heads of seabream and seabass, but these were not significant [17]. Highest antioxidant activities corresponded to frames and trimmings of both species, with the lowest values found in seabream viscera (Table 6). Antioxidant activities were comparable to those determined in FPHs from other species [12,38,40].

Similarly to antioxidant activities, antihypertensive activities reached highest % inhibition and lowest IC_50_ values in frames and trimmings (48.2%; 793.2 μg protein/mL in seabream and 50.2%; 801.3 μg protein/mL in seabass) accompanied by the opposite situation in viscera of both fish (Table 6). To our knowledge, antihypertensive activity has not been tested in seabass or seabream FPHs before, but compared to FPHs from other fish (21.5–87.0%; 165.0–1273.4 μg protein/mL) [12,38,40,42,43], the results presented here point to general low activity.

Significant positive correlations were found between Mw and both antioxidant (DPPH) and antihypertensive activities, with Pearson correlation coefficients of 0.987 and 0.938 respectively (*p* < 0.01). However, the significance of this relationship would require confirmation, as in the present study the experimental setup only includes six data points. While some works failed to find a relationship [39], others found higher DPPH activity for higher Mw peptides [44].

## 4. Conclusions

In the optimal conditions determined, both fish showed potential to produce high quality FPHs for formulation of food and food supplements. Frames and trimming appeared as the best performing material, based on the amount of protein released, antioxidant and antihypertensive activities and yield of fish oil. Nonetheless, viscera required shorter reaction times, around 1 h vs. 3 h for the other tissues, and hence less energy consumption. Furthermore, seabass viscera also contained a significant amount of oil.

The results presented here support the idea that FPHs from seabream and seabass represent a good alternative to conventional production of low-value fish meal. In particular, the proportion of pleasantly flavored and essential amino acids of these FPHs is especially appealing for its incorporation into food products. However, further work would be required to assess the performance of the products formulated with the FPHs obtained.

## Figures and Tables

**Figure 1 foods-09-01503-f001:**
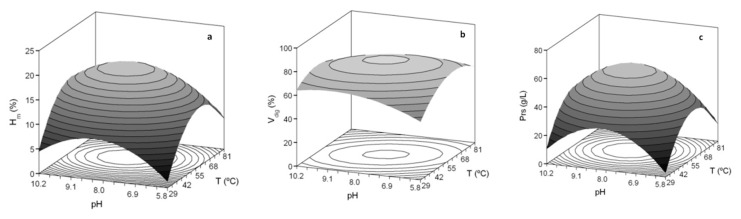
Optimization studies of Sb_He (**a**–**f**) and Sbass_He (**g**–**l**) hydrolysis by endogenous Alcalase. Theoretical response surfaces describe the joint effect of pH and *T* on *H_m_* (**a**,**d**,**g**,**l**), V_dig_ (**b**,**e**,**h**,**k**) and Prs (**c**,**f**,**i**,**l**) variables. Histograms show the individual effect of Alcalase concentration on the same variables. Error bars are the intervals of confidence for n = 2 (replicates of different hydrolysates) and α = 0.05. Different asterisks in each histogram-graph means statistically significant differences (*p* < 0.05).

**Figure 2 foods-09-01503-f002:**
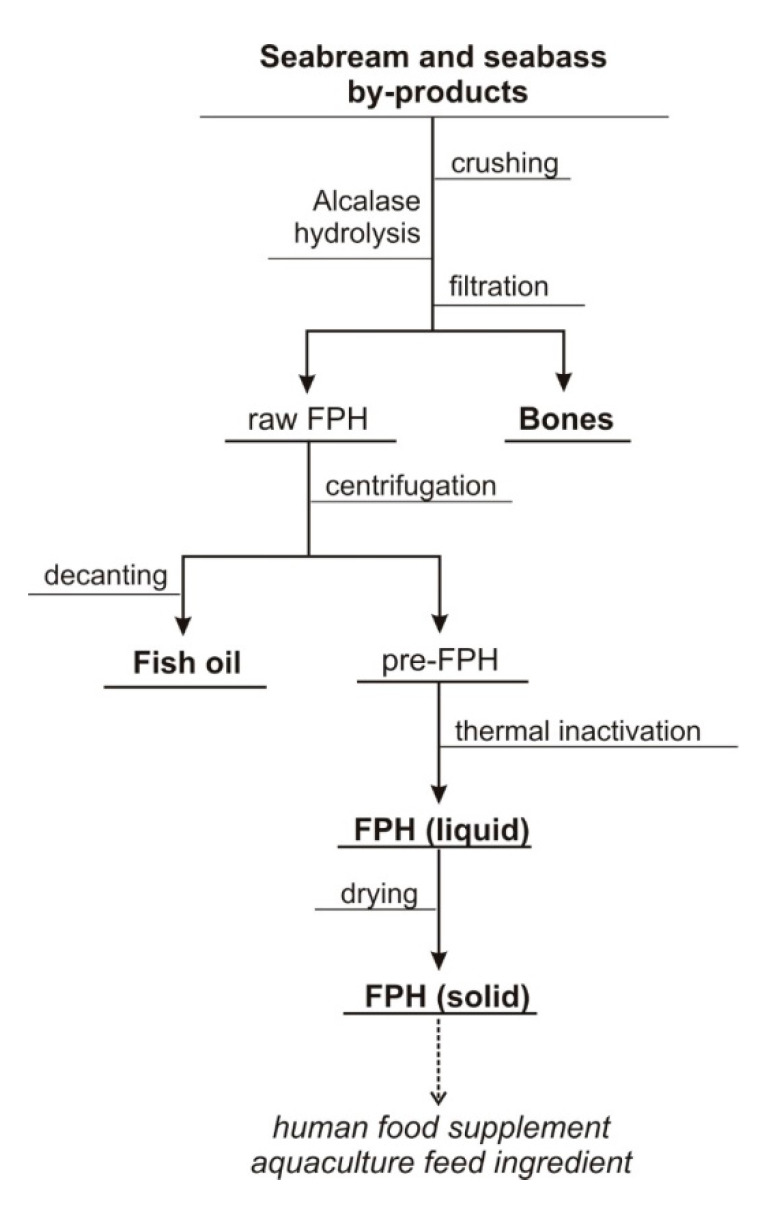
Flowchart defining the steps for the production of FPH from seabream and seabass by-products.

**Figure 3 foods-09-01503-f003:**
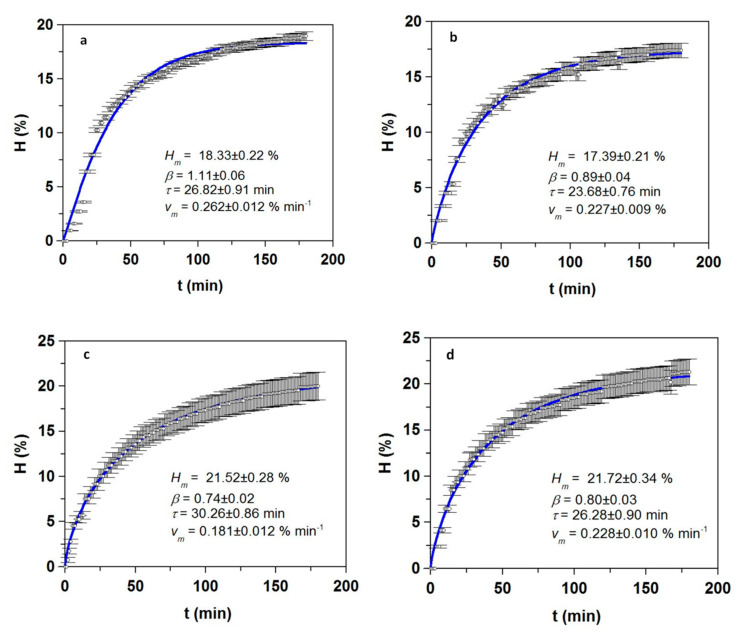
Kinetics of the hydrolysis of heads (**a**,**b**), frames and trimmings (**c**,**d**) and viscera (**e**,**f**) of seabream (**a**,**c**,**e**) and seabass (**b**,**d**,**f**). Experimental data of hydrolysis degree (symbols) were described by Equation (2) (continuous line). Error bars are the confidence intervals for n = 2 (replicates of different hydrolysates) and α = 0.05.

**Figure 4 foods-09-01503-f004:**
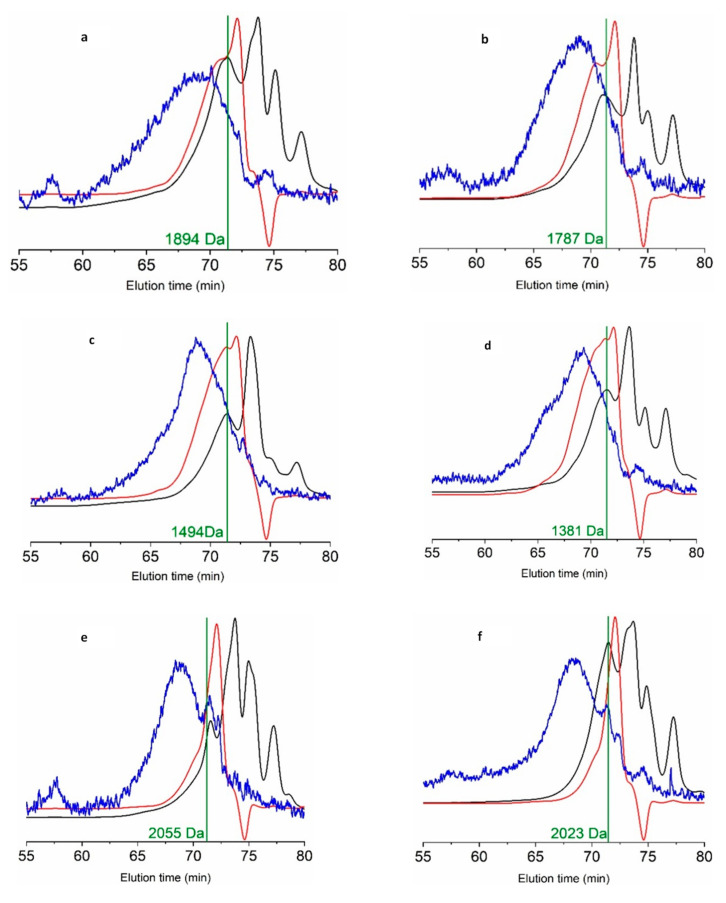
GPC eluograms of hydrolysates of heads (**a**,**b**), frames and trimmings (**c**,**d**) and viscera (**e**,**f**) of seabream (**a**,**c**,**e**) and seabass (**b**,**d**,**f**). Red line: refractive index; black line: UV (280 nm); blue line: right angle light scattering; vertical lines: weight average molecular weight (Mw).

**Figure 5 foods-09-01503-f005:**
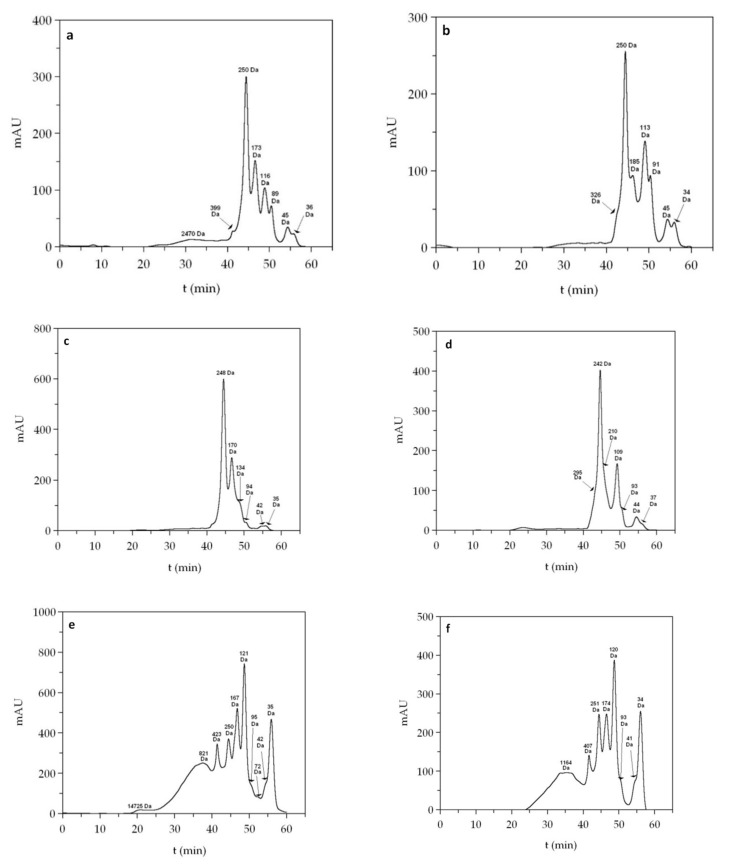
Size exclusion chromatographic profiles of heads (**a**,**b**), frames and trimmings (**c**,**d**) and viscera (**e**,**f**) of seabream (**a**,**c**,**e**) and seabass (**b**,**d**,**f**).

**Table 1 foods-09-01503-t001:** Proximate composition of Sb and Sbass by-products in terms of moisture (Mo), organic matter (OM) and ashes (Ash). Total lipids (Lip) and proteins (Pr-tN, as total nitrogen × 6.25) were determined using dried substrates. Errors are the confidence intervals for n = 2 (independent samples) and α = 0.05.

	Mo (%)	OM (%)	Ash (%)	Lip (%)	Pr-tN (%)
Sb_He	65.3 ± 1.2	28.3 ± 1.8	6.4 ± 0.5	28.5 ± 2.2	52.3 ± 2.7
Sb_FT	63.8 ± 1.9	30.6 ± 0.7	5.6 ± 0.8	40.3 ± 3.1	49.5 ± 1.8
Sb_Vis	70.2 ± 2.0	28.2 ± 1.5	1.6 ± 0.3	35.7 ± 4.1	47.2 ± 3.1
Sbass_He	63.1 ± 1.5	29.7 ± 0.7	7.2 ± 0.5	39.2 ± 2.0	49.9 ± 2.8
Sbass_FT	62.6 ± 2.0	32.2 ± 2.3	5.2 ± 0.4	44.9 ± 1.4	43.1 ± 2.0
Sbass_Vis	72.3 ± 0.9	27.2 ± 1.0	0.5 ± 0.2	54.2 ± 0.9	36.8 ± 0.8

**Table 2 foods-09-01503-t002:** Polynomial equations describing the joint effect of *pH* and temperature (*T*) on Alcalase proteolysis of Sb_He and Sbass_He. Optima values of both independent variables (*T_opt_*, *pH_opt_*) to reach the predicted maximum responses (*Y_max_*) were also calculated.

	Second Order Models	$R_{adj}^{2}$	*T_opt_* (°C)	*pH_opt_*	*Y_max_*
Sb_He	*H_m_* (%) = 18.69 + 1.64 *T* + 1.07 *pH* − 3.56 *T^2^* − 2.35 *pH^2^*	0.885	59.1	8.32	19.0%
	*V_dig_* (%) = 80.87 − 3.26 *TpH* − 6.98 *T^2^* − 3.70 *pH^2^*	0.858	55.0	8.00	80.9%
	*Prs* (g/L) = 59.69 + 3.00 *T* + 2.53 *pH* − 11.67 *T^2^* − 9.87 *pH^2^*	0.848	57.3	8.18	60.0 g/L
Sbass_He	*H_m_* (%) = 17.14 + 1.09 *T* + 1.73 *pH* − 1.79 *TpH* − 3.04 *T^2^* − 2.28 *pH^2^*	0.781	56.4	8.49	17.5%
	*V_dig_* (%) = 79.78 + 2.76 *T* + 3.16 *pH* − 2.58 *TpH* − 6.62 *T^2^* − 10.19 *pH^2^*	0.891	58.2	8.19	80.2%
	*Prs* (g/L) = 59.70 + 3.13 *T* + 7.31 *pH*− 4.84 *T^2^* − 7.41 *pH^2^*	0.871	60.7	8.70	62.0 g/L

**Table 3 foods-09-01503-t003:** Mass balances and proximal analysis of the products recovered from Alcalase hydrolysates of seabream and seabass by-products. Shown errors are the confidence intervals for n = 3 (replicates of different hydrolysates) and α = 0.05. m_b_: percentage of the bones recovered (*w*/*w* of crude substrate); V_dig_: yield of digestion process; Oil: total oil recovered (*v*/*w* of crude substrate); Prs: Total soluble protein determined by Lowry; Pr-tN: Total protein determined as total nitrogen x 6.25; TS: Total sugars; Dig: Digestibility.

FPH	m_b_ (%)	V_dig_ (%)	Oil (%)	Prs (g/L)	Pr-tN (g/L)	TS (g/L)	Dig (%)
Sb_He	19.9 ± 0.5	79.9 ± 0.8	5.9 ± 0.3	61.6 ± 1.6	63.9 ± 1.4	1.8 ± 0.2	89.6 ± 1.4
Sb_FT	14.0 ± 0.5	77.5 ± 2.0	10.6 ± 0.9	81.2 ± 3.1	83.5 ± 2.4	1.3 ± 0.1	90.7 ± 2.4
Sb_Vis	7.3 ± 2.1	89.0 ± 2.0	3.9 ± 0.2	37.9 ± 1.7	42.5 ± 1.1	0.8 ± 0.1	87.8 ± 3.9
Sbass_He	19.2 ± 1.3	79.2 ± 1.1	8.1 ± 0.4	63.3 ± 0.4	65.6 ± 2.0	1.4 ± 0.1	90.3 ± 1.5
Sbass_FT	10.6 ± 0.4	77.6 ± 0.0	13.8 ± 0.5	73.0 ± 9.5	74.9 ± 1.2	1.7 ± 0.2	91.3 ± 1.8
Sbass_Vis	-	88.6 ± 5.6	27.5 ± 2.1	33.0 ± 1.6	36.1 ± 1.9	0.9 ± 0.1	86.3 ± 2.4

**Table 4 foods-09-01503-t004:** Amino acids content of FPH from seabream and seabass by-products (% or g/100 g total amino acids). OHPro: hydroxyproline. TEAA/TAA: ratio total essential amino acids for human/total amino acids. Errors are the confidence intervals for n = 2 (replicates of independent batches) and α = 0.05.

Amino Acids	Sb_He	Sb_FT	Sb_Vis	Sbass_He	Sbass_FT	Sbass_Vis
**Asp**	9.46 ± 0.03	10.80 ± 0.01	9.52 ± 0.00	9.25 ± 0.17	9.82 ± 0.03	9.69 ± 0.16
**Thr**	4.52 ± 0.03	4.30 ± 0.07	4.63 ± 0.07	4.31 ± 0.05	4.48 ± 0.04	4.80 ± 0.03
**Ser**	4.82 ± 0.01	4.68 ± 0.04	5.07 ± 0.07	4.89 ± 0.06	4.62 ± 0.02	5.60 ± 0.03
**Glu**	13.83 ± 0.11	15.11 ± 0.11	12.91 ± 0.85	13.63 ± 0.07	14.21 ± 0.05	13.04 ± 0.21
**Gly**	10.04 ± 0.08	8.63 ± 0.17	8.57 ± 0.01	10.84 ± 0.11	9.66 ± 0.01	7.75 ± 0.04
**Ala**	7.39 ± 0.08	7.69 ± 0.15	7.23 ± 0.07	7.60 ± 0.01	7.56 ± 0.04	7.54 ± 0.01
**Cys**	0.62 ± 0.01	0.89 ± 0.01	0.97 ± 0.09	0.69 ± 0.10	0.68 ± 0.05	0.96 ± 0.16
**Val**	4.36 ± 0.01	3.70 ± 0.23	4.90 ± 0.02	4.23 ± 0.06	4.15 ± 0.03	5.54 ± 0.11
**Met**	2.95 ± 0.14	3.33 ± 0.14	2.83 ± 0.05	2.76 ± 0.04	3.04 ± 0.01	2.62 ± 0.11
**Ile**	3.38 ± 0.17	2.58 ± 0.04	3.65 ± 0.01	3.17 ± 0.04	3.40 ± 0.01	4.03 ± 0.22
**Leu**	6.65 ± 0.14	7.07 ± 0.09	7.54 ± 0.01	6.31 ± 0.06	6.72 ± 0.03	7.75 ± 0.13
**Tyr**	3.42 ± 0.02	3.59 ± 0.05	3.51 ± 0.08	3.34 ± 0.02	3.07 ± 0.02	3.99 ± 0.21
**Phe**	4.03 ± 0.07	4.64 ± 0.12	4.93 ± 0.23	4.14 ± 0.09	3.98 ± 0.01	4.96 ± 0.15
**His**	2.36 ± 0.05	2.67 ± 0.02	2.26 ± 0.01	2.07 ± 0.01	2.13 ± 0.01	2.08 ± 0.05
**Lys**	7.13 ± 0.07	7.81 ± 0.12	7.62 ± 0.06	6.93 ± 0.11	8.05 ± 0.07	7.55 ± 0.01
**Arg**	6.56 ± 0.15	5.73 ± 0.05	6.37 ± 0.06	6.39 ± 0.09	6.47 ± 0.02	5.28 ± 0.06
**OHPro**	2.70 ± 0.05	1.92 ± 0.14	2.42 ± 0.69	3.18 ± 0.13	2.31 ± 0.13	1.85 ± 0.17
**Pro**	5.79 ± 0.16	4.84 ± 0.02	5.06 ± 0.04	6.27 ± 0.10	5.66 ± 0.03	4.97 ± 0.40
**TEAA/TAA (%)**	41.94 ± 0.37	41.84 ± 0.35	44.73 ± 0.11	40.32 ± 0.29	42.42 ± 0.10	44.61 ± 0.59

**Table 5 foods-09-01503-t005:** Molecular weights and associated ranges in the distribution of peptides in the Sb and Sbass hydrolysates. PDI: polydispersity index; Mn; number average molecular weight; Mw: weight average molecular weight.

FPH	Mn (Da)	Mw (Da)	PDI	0–0.2 kDa	0.2–1 kDa	1–3 kDa	>3 kDa
Sb_He	832	1894	2.276	27.5	24.6	35.4	12.5
Sb_FT	1060	1494	1.409	18.5	23.7	51.2	6.6
Sb_Vis	790	1936	2.450	28.5	33.9	28.3	9.3
Sbass_He	1040	1787	1.718	24.7	17.9	45.9	11.5
Sbass_FT	802	1381	1.722	17.6	38.8	36.2	7.4
Sbass_Vis	1065	2023	1.900	26.3	11.3	50.2	12.2

**Table 6 foods-09-01503-t006:** In vitro antioxidant and antihypertensive activities of FPH from Sb and Sbass waste. Mean ± confidence intervals for n = 2 (samples from independent hydrolysates) and α = 0.05. *I_ACE_*: inhibitory activity of angiotensin I converting enzyme (ACE) and *IC*_50_: half maximal inhibitory concentration.

	ANTIOXIDANT			ANTIHYPERTENSIVE	
FPH	DPPH (%)	ABTS (μg BHT/mL)	Crocin (μg Trolox/mL)	*I_ACE_* (%)	*IC*_50_ (μg protein/mL)
Sb_He	43.2 ± 2.2	11.12 ± 0.98	6.12 ± 0.34	43.1 ± 3.2	1034.5 ± 145.4
Sb_FT	52.4 ± 2.5	15.09 ± 1.78	7.39 ± 0.67	48.2 ± 2.9	793.2 ± 127.1
Sb_Vis	37.4 ± 1.5	9.84 ± 0.45	4.98 ± 1.59	37.0 ± 3.4	1245.8 ± 76.3
Sbass_He	45.0 ± 3.2	11.89 ± 0.46	5.78 ± 0.73	40.5 ± 1.5	989.2 ± 68.0
Sbass_FT	53.9 ± 1.5	14.46 ± 1.43	6.88 ± 0.82	50.2 ± 2.5	801.3 ± 56.4
Sbass_Vis	41.1 ± 1.8	10.76 ± 0.75	5.15 ± 0.91	33.8 ± 5.4	1398.3 ± 89.6

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
