# Peer review of "Optimization of the Enzymatic Protein Hydrolysis of By-Products from Seabream (*Sparus aurata*) and Seabass (*Dicentrarchus labrax*), Chemical and Functional Characterization"

_foods, 2020, doi:10.3390/foods9101503_

Round 1

Reviewer 1 Report

Article revision: Optimization of the enzymatic protein hydrolysis of by-products from seabream (Sparus aurata) and seabass (Dicentrarchus labrax), chemical and functional characterization

General comments:

The manuscript describes the valorization of fish by enzymatic hydrolysis of by products from seabream and seabass. The idea beyond the work is worthy and the results are presented in a clear and accurate manner.

Specific comments:

  • The authors mention the valorisation of the fish by products emphasising that this by-products should not be discarded however they only valorise a part f these by-products, what is done to the rest? Is it discarded?
  • Considering the high percentage of lipids, is not the action of an hydrolytic enzyme compromised? If the aim of the work is to obtain protein hydrolysates, wouldn’t be more efficient to remove first the lipids and then apply the enzyme for proteolytic action? The enzymatic action in such heterogeneous material can be hindered, and a pre-treatment of the material would allow to decrease the final time of processing.

Author Response

Response to  Reviewer 1 comments

The authors would like to thank the reviewer for his comments and suggestions. Answers to specific points risen follow:

General comments:

The manuscript describes the valorization of fish by enzymatic hydrolysis of by products from seabream and seabass. The idea beyond the work is worthy and the results are presented in a clear and accurate manner.

 Specific comments:

The authors mention the valorisation of the fish by products emphasising that this by-products should not be discarded however they only valorise a part f these by-products, what is done to the rest? Is it discarded?

ANSWER: In the present manuscript, we have worked with all the by-products commonly generated from the food processing of seabream and seabass (viscera, heads, trimmings and frames) to obtain clean fillets. From the proteolysis process proposed, we have obtained fish oil, fish protein hydrolysates (FPHs) and clean bones. Fish oils from seabream and seabass did not present an interesting omega-3 composition for its use in high-added value compounds, but they could be employed in the industrial production of dyes and paints. The characteristics and properties of FPHs were remarkable and they are adequate candidates to be employed as ingredients in foods and also in aquaculture diets, evidently, not to feed specimens of seabream nor seabass. At this moment, we are studying if the chemical composition of bones would allow their incorporation into nutraceutical supplements or in fertilizers. Besides, if the presence of collagen in bones is sufficiently large, an optimal protocol for its recovery should be developed. Both studies exceed the objetives of this report.

Considering the high percentage of lipids, is not the action of an hydrolytic enzyme compromised? If the aim of the work is to obtain protein hydrolysates, wouldn’t be more efficient to remove first the lipids and then apply the enzyme for proteolytic action? The enzymatic action in such heterogeneous material can be hindered, and a pre-treatment of the material would allow to decrease the final time of processing.

ANSWER: Thank you so much for your valuable comment. The previous extraction of oils can be performed using organic solvents (diethyl ether, methanol, chloroform, etc.). However, our sustainable and environmentally friendly approach try to avoid the use of pollutant solvents. In addition, and from an industrial viewpoint, the cost of such solvents, the solid:liquid ratio needed for oil extraction and the difficulty generated by the management of the residual effluents that thsesnsolvents produce, makes their application unviable.

A more environmentally friendly alternative to recover oils can be the use of thermo-mechanical extrusion, but unfortunately we do possess the adequate equipment to study this strategy. In any case, a thermal process of these characteristics (70-90°C) would also imply unwanted changes in the substrate to be hydrolyzed.

Regarding our presented results, we have obtained an optimal process to valorise seabream and seabass wastes (still unexplored) recovering oils, bones and producing FPHs.  It is possible that the proteolysis step could be shortened without the presence of oils, but during the three hours of hydrolysis this run well (see kinetics of hydrolysis degree and the digestion values as Vdig), the composition of FPHs was excellent (in terms of amino acids, soluble protein, etc.), and the functionality (in vitro digestibility, bioactives) was also remarkable.

Reviewer 2 Report

The manuscript is a scientific novelty. The assumptions of the work are well formulated. The methodology is well described. The authors made a proper statistical analysis of the obtained research results. The authors made an in-depth discussion.

A positive evaluation of the manuscript does not mean that the work cannot be improved. Namely, the authors should present in the supplementary materials all the obtained GC chromatograms from the fatty acid analysis.

Author Response

Response to  Reviewer 2 comments

The authors would like to thank the reviewer for his comments and suggestions. Answers to specific points risen follow:

The manuscript is a scientific novelty. The assumptions of the work are well formulated. The methodology is well described. The authors made a proper statistical analysis of the obtained research results. The authors made an in-depth discussion.

A positive evaluation of the manuscript does not mean that the work cannot be improved. Namely, the authors should present in the supplementary materials all the obtained GC chromatograms from the fatty acid analysis.

ANSWER: An extra figure (S1) has been added to the supplementary material displaying the GC-MS chromatograms as suggested by the reviewer.

Reviewer 3 Report

Manuscript foods-958962 is an interesting and novel topic, concerning the exploitation of fish protein hydrolysates from different fish species, as food supplements or formulation of other foods. The statistical analysis supports well the data obtained. However, my main concern is the numerous grammar errors through the text and the consecutive use of the first person (i.e., we...). In my opinion, the authors should seek the help of a native English speaker to correct language errors. Other specific comments are as follows:

-Abstract

Line 24.Change ''range'' to ''ranged''.

Line 26.''reached'',''displayed''.

-Materials and Methods

Line 69. were grounded?  Change ''ground''.

Line 103. There is a typing error in the symbol within the parenthesis.

-Results and Discussion

Lines 168-175 and elsewhere. Kindly try to modify the sentences. Avoid the use of the first person.

Line 212.'' statistically significant differences''.

Line 213.'' allowed''.

Line 293.''reached''.

Line 304. ''predominated''.

Line 314. ''showed''.

Line 340.''contained''.

Based on the aforementioned,I suggest a major revision of the present article.

Author Response

Response to  Reviewer 3 comments

The authors would like to thank the reviewer for his comments and suggestions. Answers to specific points risen follow:

Manuscript foods-958962 is an interesting and novel topic, concerning the exploitation of fish protein hydrolysates from different fish species, as food supplements or formulation of other foods. The statistical analysis supports well the data obtained. However, my main concern is the numerous grammar errors through the text and the consecutive use of the first person (i.e., we...). In my opinion, the authors should seek the help of a native English speaker to correct language errors. Other specific comments are as follows:

-Abstract

Line 24.Change ''range'' to ''ranged''.

Line 26.''reached'',''displayed''.

-Materials and Methods

Line 69. were grounded?  Change ''ground''.

Line 103. There is a typing error in the symbol within the parenthesis.

-Results and Discussion

Lines 168-175 and elsewhere. Kindly try to modify the sentences. Avoid the use of the first person.

Line 212.'' statistically significant differences''.

Line 213.'' allowed''.

Line 293.''reached''.

Line 304. ''predominated''.

Line 314. ''showed''.

Line 340.''contained''.

Based on the aforementioned,I suggest a major revision of the present article.

ANSWER: All the above changes have been made in the manuscript, except replacement of "ground" to "grounded" in line 69, as "ground" is the correct past participle form of the verb "grind".  Also, "reach" was not found in line 293 but other present forms of this verb where changed to the past where suitable. In this line, all present tense forms were changed to the past form throughout the document for consistency. As suggested by the reviewer, to avoid the use of the first person, the sentences concerned were changed to the passive voice. The document was thoroughly proof read and grammar errors and typos have been corrected.

Reviewer 4 Report

The paper entitled "Optimization of the enzymatic protein hydrolysis of by-products from seabream (Sparus aurata) and seabass (Dicentrarchus labrax), chemical and functional characterization" presents a fairly serious and rigorous study. The authors have made a great work, taking into account many experimental variables that influence in the enzymatic protein hydrolysis. Both the experimental part (chemical analysis), and the part of optimization of variables by means of design of experiments have been well raised.

Author Response

Response to  Reviewer 4 comments

The authors would like to thank the reviewer for his time in reviewing the present manuscript.

The paper entitled "Optimization of the enzymatic protein hydrolysis of by-products from seabream (Sparus aurata) and seabass (Dicentrarchus labrax), chemical and functional characterization" presents a fairly serious and rigorous study. The authors have made a great work, taking into account many experimental variables that influence in the enzymatic protein hydrolysis. Both the experimental part (chemical analysis), and the part of optimization of variables by means of design of experiments have been well raised.

Round 2

Reviewer 3 Report

The authors have responded adequately to my suggestions. Therefore, I suggest the publication of their revised article.